



# Robust increase of Indian monsoon rainfall and its variability under future warming in CMIP-6 models

Anja Katzenberger[1,2], Jacob Schewe[1], Julia Pongratz[2,3], and Anders Levermann[*1,4,5]

[1]Potsdam Institute for Climate Impact Research, Potsdam, Germany
[2]Ludwig-Maximilian University, Munich, Germany
[3]Max Planck Institute for Meteorology, Hamburg, Germany
[4]LDEO, Columbia University, New York, USA
[5]Potsdam University, Potsdam, Germany

**Correspondence:** [*]Anders Levermann (anders.levermann@pik-potsdam.de)

**Abstract.** The Indian summer monsoon is an integral part of the global climate system. As its seasonal rainfall plays a crucial role in India's agriculture and shapes many other aspects of life, it affects the livelihood of a fifth of the world's population. It is therefore highly relevant to assess its change under potential future climate change. Global climate models within the Coupled Model Intercomparison Project Phase 5 (CMIP-5) indicated a consistent increase in monsoon rainfall and its variability under

global warming. Since the range of the results of CMIP-5 was still large and the confidence in the models was limited due to partly poor representation of observed rainfall, the updates within the latest generation of climate models in CMIP-6 are of interest. Here, we analyse 32 models of the latest CMIP-6 exercise with regard to their annual mean monsoon rainfall and its variability. All of these models show a substantial increase in June-to-September (JJAS) mean rainfall under unabated climate change (SSP5-8.5) and most do also for the other three Shared Socioeconomic Pathways analyzed (SSP1-2.6, SSP2-

4.5, SSP3-7.0). Moreover, the simulation ensemble indicates a linear dependence of rainfall on global mean temperature with high agreement between the models and independent of the SSP; the multi-model mean for JJAS projects an increase of 0.33 mm/day and 5.3% per degree of global warming. This is significantly higher than in the CMIP-5 projections. Most models project that the increase will contribute to the precipitation especially in the Himalaya region and to the northeast of the Bay of Bengal, as well as the west coast of India. Interannual variability is found to be increasing in the higher-warming scenarios

by almost all models. The CMIP-6 simulations largely confirm the findings from CMIP-5 models, but show an increased robustness across models with reduced uncertainties and updated magnitudes towards a stronger increase in monsoon rainfall.

## 1   Introduction

As one of the integral components of the global climate system, the Indian monsoon provides water to the densely populated region of South Asia. About 80% of the annual precipitation over India occurs during the summer period supplying water to

the crops during the prime agricultural season (Bollasina, 2014). The crop yields (especially rice is dominant in the region) are highly sensitive to the monsoon rainfall variability (Prasanna, 2014; DeFries et al., 2016). As agriculture contributes to about 20% of the gross domestic product (Zaveri et al., 2016), the monsoon's rainfall also has a retractable effect on India's economy





(Gadgil and Gadgil, 2006). Therefore, there is an inextricable link between the Indian summer monsoon and the health as well as the socio-economic well-being of people. Thus, understanding the response of the Indian summer monsoon and its interannual variability to different global warming scenarios is critical for designing management strategies of water resources and agricultural policies in the future.

Observations of the Indian summer monsoon in central India have revealed a decreasing rainfall trend in the second half of the 20th century (Ramanathan et al., 2005; Bollasina et al., 2011; Mishra et al., 2012, 2014b; Shah and Mishra, 2016; Jin and Wang, 2017). Since the convergence of the monsoon low-level circulation peaks in central India and effects of subseasonal und interannual variability affect the region in the largest extent, central India is often used as an indicator for the Indian monsoon (Singh et al., 2019). The declining trend has reversed in various datasets since the beginning of the 21st century except in the Indian Meteorological Department dataset where a stabilization was captured (Jin and Wang, 2017). Sinha et al. (2015) analysed the monsoon oscillation over the past two millennia and found that the drying trend since the 1950s is within but on the very edge of the historic envelope of natural variability.

Multi-millennial paleorecords indicate strong changes both in the Indian and East Asian summer monsoon (Wang et al., 2005b, a, 2008; Zhang et al., 2008; Li et al., 2017; Wang et al., 2017; Zhang et al., 2019; Ming et al.; Wang et al., 2020). While it is speculated (Schewe et al., 2012; Herzschuh et al., 2014; Wang et al., 2020), that there might be abrupt monsoon changes due to a moisture-advection feedback at play (Levermann et al., 2009), these are generally associated with either aerosol forcing or changes in the sea surface temperatures of the surrounding ocean waters. Under future warming an overall strengthening of the monsoon rainfall is expected due to enhanced atmospheric moisture bearing capacity.

Several studies associate the weakening trend of the Indian monsoon with the warming of the Indian Ocean sea surface temperature and the relatively dampened warming over the Indian subcontinent (Zhou et al., 2008; Deser et al., 2010). The resulting decrease in the land-sea thermal gradient over South Asia and the consequently subdued Hadley circulation have lead to a reduction of the rainfall amount during the summer period since the 1950s (Roxy et al., 2015).

The ocean warming trend has been attributed to both natural and anthropogenic causes, but model results indicate a clear responsibility in the latter accounting for 98.8% of the external forcing trend (Dong et al., 2014). Changes in the Pacific Ocean sea surface temperature associated with El Niño Southern oscillation (ENSO) might have contributed to the Indian Ocean warming trend in the second half of the 20th century (Rasmusson and Carpenter, 1983; Roxy et al., 2014). As changes in the Walker circulation due to ENSO dampen the monsoonal winds and suppress rainfall over the subcontinent, El-Niño events typically coincide with dry monsoon years, while La-Niña years are often accompanied by strong monsoon rainfall (Kumar et al., 2006). Therefore, the accumulation of El Niño years in the second half of the 20th century might have contributed to the warming pattern of the Indian Ocean and the drying of the Indian monsoon over land (Roxy et al., 2014; Singh et al., 2019).

That the land warmed less relative to the ocean is attributable to the high aerosol emissions of the region: Anthropogenic emission including sulfate aerosols rose steeply in India and neighboring regions since the 1950s due to the strong expansion of industry and the population growth (Acharya and Sreekesh, 2013; Krishna Moorthy et al., 2013). Aerosols directly affect the incoming solar radiation by scattering and absorbing processes contributing to a decrease of the meridional thermal contrast (Li et al., 2016). Further, aerosols increase cloud fraction, lead to more but smaller droplets and thus increase cloud albedo





(Twomey, 1977; Guo et al., 2015). These indirect effects via aerosol-cloud interactions play the predominating role of the weakening monsoon circulation (Bollasina et al., 2011; Guo et al., 2015; Sarangi et al., 2018). As seen in the second half of the
20th century, aerosols have the potential to offset the competing impact of GHG over South Asia (Guo et al., 2015). While the general role of aerosols is widely agreed on, the magnitude of aerosol impacts remain highly uncertain (Rosenfeld et al., 2014). Challenges result from the lack of long-term observations of aerosols and the difficulty to represent the underlying physical processes in models (Singh et al., 2019).

While the effects of atmospheric constituents have been intensely studied, less attention has been dedicated to the role
of widespread agricultural expansion and intensification (Singh et al., 2019). Recent studies suggest that land-use changes contributed to the weakening of the monsoon through various effects. Krishnan et al. (2016) found that the deforestation associated with agricultural expansion contributed to an increase of the planetary albedo and thus the weakening of the monsoon meridional overturning circulation in the second half of the 20th century. Paul et al. (2016) links the deforestation to an diminished evapotranspiration and convection which dampens recycling of precipitation for the monsoon rainfall. Over some
parts of India the share of recycled precipitation to mid- and late monsoon rainfall can exceed 25% (Pathak et al., 2014). Barlow et al. (2016) points out that the importance of the vegetation feedback may raise under warming conditions. Other studies found a regional statistically significant effect of irrigation which through its surface cooling effect again may have contributed to the reduction of the land-sea thermal gradient (Lee et al., 2009; Shukla et al., 2014; Cook et al., 2015; de Vrese et al., 2016). The magnitudes of these effects in combination with other forcing have still to be clarified since most studies focus on the isolated
effect of land cover change or single model results (Singh et al., 2019).

The revival of the rainfall in central and north India since 2002 might be explained by the emerging land warming due to GHG emissions. The raising land surface temperature increases the meridional temperature gradient in the lower troposphere, enhancing the Hadley circulation and summer monsoon rainfall (Jin and Wang, 2017). Therefore, the magnitude of future monsoon rainfall may depend on where temperature rises faster – on the sea surface or land masses (Singh et al., 2019). Since
this goes back to the competing influence of GHG and aerosol forcing over land, the task of modeling the future monsoon rainfall coincides with projecting the magnitude of the different forcing mechanisms and capturing the monsoon's sensitivity to it.

The capabilities of climate models in simulating the Indian monsoon have improved over time, such that more accurate projections can be expected from the latest update of the climate models in CMIP-6. Several studies found a broad range of
improvements between CMIP-3 and CMIP-5 in simulating the 20th century monsoon (Sperber et al., 2013; Ogata et al., 2014; Ramesh and Goswami, 2014) though one study, based on a small subset of models, however, disagrees (Shashikanth et al., 2014). Gusain et al. (2020) found a significant improvement between CMIP-5 and CMIP-6 in simulating the Indian summer monsoon rainfall for the period 1951-2005. Models in CMIP-5 still struggled with various issues including displaying the decrease in rainfall in the second half of the 20th century (Saha et al., 2014; Sabeerali et al., 2015; Ashfaq et al., 2017), capturing
observed trends in the extremes (Mishra et al., 2014a) and seasonality indices (Ul Hasson et al., 2016). With the new generation, models' capacities in capturing the spatiotemporal pattern of Indian summer monsoon, especially in the Western Ghats and the North-East foothills of Himalaya mountains, have undergone significant progress (Gusain et al., 2020). While global coupled





models in CMIP-5 failed to capture the influence of topography, land-surface-feedback and land use change due to their coarse

spatial resolution, the general higher resolution in CMIP-6 contributes to an improved simulation of Indian monsoon dynamics

(Singh et al., 2019; Gusain et al., 2020). Further improvements have been achieved by updating deep convective schemes,

modifying parameterization on microphysical scale, integrating indirect effects of aerosols in cloud formation and advancing

ocean-ice models (Gusain et al., 2020).

Within the latest studies using global coupled models, there is a widespread consensus that the Indian monsoon rainfall will

increase due to climate change in the 21st century (Chaturvedi et al., 2012; Menon et al., 2013; Lee and Wang, 2014; Asharaf

and Ahrens, 2015; Mei et al., 2015; Sharmila et al., 2015; Varghese et al., 2020). This trend is found for various CMIP-5

models (Menon et al., 2013), the multi-model mean (Chaturvedi et al., 2012), the mean of onle the four best models (Lee and

Wang, 2014) or the model with the best deep convection scheme (Varghese et al., 2020). Under Representative Concentration

Pathway 8.5 (RCP-8.5) CMIP-5 models project a median increase in Indian monsoons rainfall of 2.3%/K (Menon et al., 2013).

Also under SSP5-8.5, the amount of rainfall over India is projected to increase by 18.7% by the end of the 21st century

compared to 1961-1999 (Chaturvedi et al., 2012). This trend is expected to be the consequence of the warming of the Indian

Ocean enhancing atmospheric moisture content and thus moisture convergence (Cherchi et al., 2011; Seth et al., 2013; Mei

et al., 2015; Sooraj et al., 2015). This so called thermodynamic effect dominates over the dynamic effect which refers to a

reduced monsoon circulation due to a weakened walker circulation and an expected decrease of rainfall (Vecchi et al., 2006;

Mei et al., 2015; Sooraj et al., 2015). The interannual variability is projected to increase in most models under the strongly

forced scenarios as well as in models with good performance in capturing the mean seasonal cycle in the present climate (Kitoh

et al., 1997; Menon et al., 2013; Jayasankar et al., 2015; Sharmila et al., 2015; Kitoh, 2017). The uncertain role of

Here, we aim to update the CMIP projections for the Indian monsoon rainfall and its interannual variability for the 21st

century by using 32 models of the latest climate model generation. For this purpose, we use the shared socioeconomic pathways

and possible corresponding forcing levels as scenario framework (O'Neill et al., 2017). Section 2 gives a brief overview of the

data used and processed. In section 3.1 we evaluate the participating models according to their capacity of modeling the Indian

summer monsoon in historic periods. Section 3.2 presents the results of the mean summer monsoon precipitation while section

3.3 focuses on the long-term trend of interannual variability. The results are discussed in section 4.

## 2 Data and methods

In this study, we investigate the mean Indian summer monsoon rainfall and its interannual variability under four different

scenarios using 32 global climate models that participated in CMIP-6. The four scenarios (SSP1-2.6, SSP2-4.5, SSP3-7.0 and

SSP5-8.5) are based on different socio-economic scenarios and combined with the resultant forcing level (Van Vuuren et al.,

2014; O'Neill et al., 2017). The models are chosen according to their data availability for the historic period (1850-2015) and

the future period (2015-2100) under SSP5-8.5 in ScenarioMIP (Tebaldi et al., 2020). For each model, for consistency, we use

one ensemble member (if available: r1i1p1f1) even if more are available. An overview of modeling centers and data availability

for the different scenarios is given in Table 1. The short names of the models used in this study can be found in Table 2. We





select the land area with longitude 67.5°0'0"E - 98°0'0"E and latitude 6°0'0"N-36°0'0"N, comprising India and neighboring regions. The land area is obtained by using land-sea-masks for each model that are based on the percentage of the grid cells occupied by land (see Fig. 3 for each model). The resolution strongly differs between the models ranging over land from about 100 km to 500 km (See Table 2). Mean rainfall is obtained by averaging the monthly rainfall data from June-September over the

region of interest. The WFDE5 dataset of precipitation over land (Cucchi et al., 2020) is used to evaluate the models capacity of representing the Indian monsoon. This dataset has been generated on the basis of ERA5 reanalysis data and has undergone bias-adjustment methods following Weedon et al. (2010, 2011). It is provided at 0.5° spatial resolution. For calculating the change in interannual variability, we apply the Singular Spectrum Analysis Method (Golyandina and Zhigljavsky, 2013) with a window size of 20 years to extract the nonlinear trend.

## 3   Results

### 3.1   Model comparison

To evaluate the models' quantitative capacities of capturing the Indian monsoon rainfall, we compare their projected seasonal mean rainfall with WFDE5 reanalysis data over land (Cucchi et al., 2020) for two 30-year-periods in the past (1900-1930, 1985-2015). We choose these periods to obtain a model evaluation for a historic period as well as for a period close to present.

The seasonal mean rainfall from the reanalysis data is 6.1 mm/day with a standard deviation of 0.5 mm/day for 1900-1930 and 6.1 ± 0.4 mm/day for 1985-2015 (Fig. 1). For both periods, about half of the models capture the quantitative JJAS rainfall within twice the standard deviation (dashed lines in Fig. 1). The standard deviation of the models ranges from 0.3 to 0.8 mm/day for 1900-1930 and 0.3 to 1.0 mm/day in 1985-2015 (error bars in Fig. 1). The models INM-CM4-8 and FIO-ESM-2-0 overestimate the annual rainfall for both periods, the mean value of BCC-CSM2-MR exceeds the upper threshold in 1985-2015.

Several models underestimate the seasonal mean rainfall, especially the models of the Canadian Centre for Climate Modeling and Analysis (CanESM5-CanOE, CanESM5) which capture just about half of the reanalysis rainfall amount. All models that underestimate the rainfall for 1900-1930 show rainfall means below the lower threshold in 1985-2015, too. GFDL-CM4 for 1900-1930 and GISS-E2-1-G for 1985-2015 capture the seasonal rainfall quantitatively best. The other models that are closest to the reanalysis mean overlap for both periods, e.g. CNRM-CM6-1, NorESM2-MM and FGOALS-f3-L. For the two chosen

time periods, models that capture, over- or underestimate the mean rainfall within twice the standard deviation mostly have the same tendency for both periods. The multi-model mean for 1900-1930 is 5.6 ± 1.1 mm/day and 5.7 ± 1.1 mm/day for 1985-2015.

   In order to identify models with a potentially realistic representation of the Indian monsoon rainfall, we also analyze the spatial precipitation distribution for 1985-2015. We choose this period since it is closer to present time and therefore closer

to the simulated time period in the future. As reference data set we use WFDE5 reanalysis data again. The distribution is dominated by rainfall over the Western Ghats, the Himalaya region, the west coast of the Bay of Bengal, the northeast of India and the north of Myanmar partly even exceeding 20 mm/day averaged over JJAS and the 30 year period . The east of central India reaches rainfall values above 10 mm/day (Fig. 2). The spatial rainfall pattern for the CMIP-6 models in 1985-2015 is





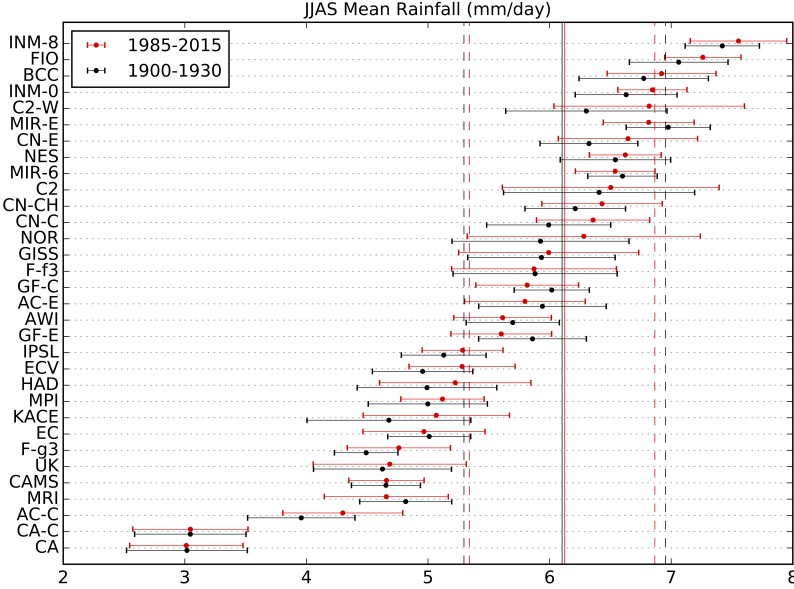

**Figure 1.** Indian summer monsoon mean rainfall (mm/day) over the region displayed in Fig.3 from 32 different models for the period 1985-2015 (red) and 1900-1930 (black). The vertical line represents the mean monsoon rainfall from WFDE5 reanalysis data for the same periods, the dashed lines show plus/minus twice the standard deviation across the 30-year time period. Circles with error bars represent mean and mean plus/minus one standard deviation for each CMIP-6 model in the same region and the same period.

shown in Fig.3. Models that captured the rainfall quantitatively well mostly simulate a spatial pattern close to the reference

distribution e.g. NorESM2-MM, CNRM-CM6-1, FGOALS-f3-L. FIO-ESM-2-0 overestimates the rainfall in the Himalaya region. The models with the tendency to underestimate the rainfall as ACCESS-CM2, CanESM5-CanOE, CanESM5 mostly are not able to capture the spatial pattern. Especially the southwest coast of India and the Himalaya region are not reproduced according to the reanalysis data by most of these models. An exemption for the models with low rainfall values are the models of EC-Earth-Consortium (EC-Earth3, EC-Earth3-Veg) which simulate a pattern very close to the reference distribution.

Presenting and discussing the results of this study, we decided to focus on the models within mean plus/minus twice the standard deviation which also deliver a reasonable spatial rainfall pattern. Nevertheless, we will provide information for all 32 models.

## 3.2    Trend in Indian summer monsoon mean rainfall for the end of the 21st century

In order to determine the long-term trend in Indian monsoon rainfall, we first analyze the temporal time series between 1850-

2100 for all models under SSP5-8.5 (Fig. 4). All available models show a clear positive long-term trend. The models exceed the envelope of the baselines variability (grey vertical lines in Fig. 4) between 2014 (HadGEM3-GC31-LL) and 2088 (CESM2),




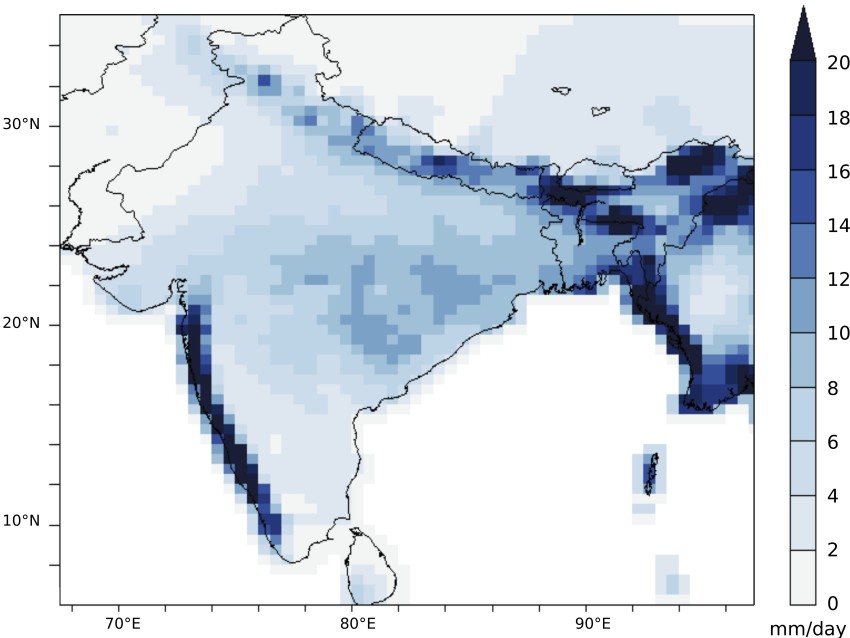

**Figure 2.** Spatial distribution of Indian summer monsoon rainfall (mm/day) averaged over the period 1985-2015 from WFDE5 reanalysis data.

on average over all models in 2045. For the other SSPs, the evolution in time as well as the magnitude of change by the end of the 21st century is indicated as model mean in Fig. 5.

To analyze the change in mean seasonal rainfall until the end of the 21st century, we calculate the difference between the
periods 2070-2100 and 1985-2015 for the four SSPs. In the stronger forced scenarios (SSP3-7.0 and SSP5-8.5), all models project an increase of precipitation. In the scenarios with less forcing (SSP1-2.6 and SSP2-4.5), the clear majority of models project an increasing trend, too. The only models to project a decrease are the models of the National Center for Atmospheric Research (CESM2-WACCM in SSP1-2.5 and SSP2-4.5 and CESM2 in SSP2-4.5). On average over all models an increase of 24,3% is projected under SSP5-8.5 (Fig. 6) and of +18,6% in SSP3-7.0 (Appendix Fig. A1), of +11,9% in SSP2-4.5 (Appendix
Fig. B1) and of +9.7% in SSP1-2.6 (Appendix Fig. C1). CanESM5 and CanESM5-CanOE show the maximum relative increase in all scenarios by the end of the 21st century. But as shown in Fig. 1 and Fig. 3, they clearly underestimate the rainfall and do not capture a realistic pattern of the rainfall distribution. CESM2-WACCM shows the minimal increase of 7.8% under SSP5-8.5. This model was able to capture the mean rainfall in 1985-2015 within twice the standard deviation and is able to capture a reasonable pattern of the rainfall. Focusing on the models that captured the mean rainfall in 1985-2015 within twice
the standard deviation (upper panel in Fig. 6), the relative increase is 17.4% under SSP5-8.5, i.e. slightly less than the average over all models. Also in the other scenarios the trend is less for these models compared to the average over all models. In summary, a robust increase of seasonal rainfall between 1985-2015 and 2070-2100 can be derived under global warming.



**Figure 3.** Spatial distribution of Indian summer monsoon mean rainfall (mm/day) averaged over the period 1985-2015. The models are shown in the same order as in Fig. 1.





**Figure 3 (continued).**

Most models project that this increase will contribute to the precipitation especially in the Himalaya region and to the northeast of the Bay of Bengal, as well as the Western Ghats (Fig. 7). Individual models indicate decreasing rainfall along the southwest coast of India and around Myanmar.

Furthermore, we analyzed the dependence of rainfall on global mean temperature (GMT, Fig. 8). The simulation ensemble indicates a linear dependence of rainfall on GMT, with a high agreement between models and independent of the scenarios. The multi-model mean indicates an increase of 0.33 mm/day ranging from 0.11 mm/day to 0.54 mm/day. The relative dependence is 5.3% per degree of global warming ranging from 1.7%/K to 13.4%/K for SSP5-8.5 across models. Considering only the more realistic models, the projected mean change is 6.1%/K for SSP5-8.5.

**Figure 4.** Time series of Indian summer monsoon mean rainfall (mm/day) for the period 1850-2100 from the 32 climate models under SSP5-8.5. The underlying area is according to the displayed region in Fig. 3. Red shadings represent the yearly values, red lines represent the nonlinear trend obtained from a Singular Spectrum Analysis with a window size of 20 years according to the method from Golyandina and Zhigljavsky (2013). The horizontal black lines represent mean plus/minus standard deviation for each model for the period 1850-2015. The order is according to Fig 1.



**Figure 4 (continued).**

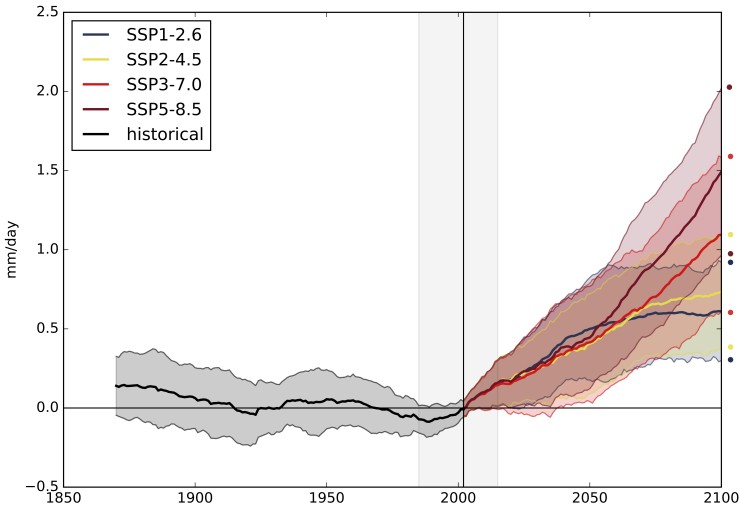

**Figure 5.** Multi-model mean of Indian summer monsoon rainfall (mm/day) over displayed area in Fig. 3 for 1870-2100 relative to the mean (horizontal black line) in 1985-2015 (grey background) for the four scenarios SSP1-2.6, SSP2-4.5, SSP3-7.0 and SSP5-8.5. The 20 years backwards smoothed time series of one ensemble member per model was used to calculate the multi-model mean. Shading in the time series represents the range of mean plus/minus one standard deviation marked with circles on the right sight of the Figure. Availability of the models according to Table 1.

### 3.3 Long-term trend of interannual variability

In order to analyze the future evolution of interannual variability, we removed the nonlinear trend obtained by a Singular Spectrum Analysis from the rainfall data as displayed in Fig. 4, and use the percentage changes in standard deviation for the period 2050-2100 with respect to 1900-1950. Under SSP5-8.5, 28 of 32 models indicate an increase of interannual variability

(Fig. 9); the multi-model mean in this scenario indicates an increase of 21.3%. The strongest increase of 56.2% is simulated by EC-Earth3-Veg which is a model that does not capture the quantitative rainfall of the Indian summer monsoon well. Four models simulate a decrease in SSP5-8.5: Both models from INM (INM-CM4-8, INM-CM5-0) and two models from CNRM-CERFACS (CNRM-CM6-1-HR, CNRM-ESM2-1) project a decrease in interannual variability. Even if two of the four models projecting a decrease under SSP5-8.5 show a relatively small decrease of less than 5%, it has to be noted that all of this four but INM-CM4-8 captured the rainfall in 1985-2015 within twice the standard deviation, making them more reliable in

four but INM-CM4-8 captured the rainfall in 1985-2015 within twice the standard deviation, making them more reliable in projecting the Indian summer monsoon than some other models. Nevertheless, among the 16 models within twice the standard deviation, 13 project an increase in interannual variability. In SSP3-7.0, 22 out of the available 27 models project an increase of interannual variability (See Appendix Fig E1). The signal in the scenarios with less forcing is less clear (See Appendix Fig. F1 and Fig. G1), but even in SSP1-2.6 still 21 out of 31 available models project an increase in interannual variability

until the second half of the 21st century. For the purpose of comparison, we also calculated the change without removing the trend and found that for SSP5-8.5 all models project an increase in interannual variability in average 39.9%. Fig. 10 shows the





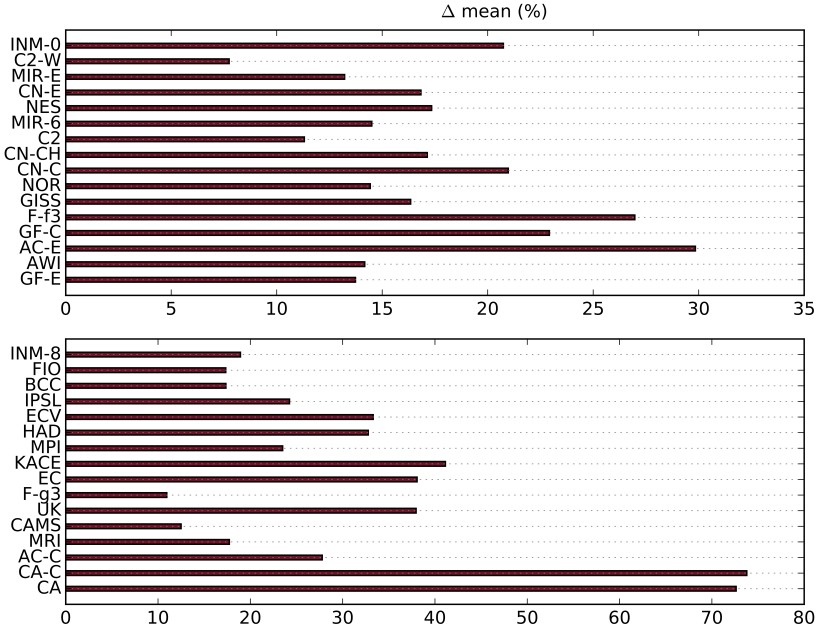

**Figure 6.** Percentage change in Indian summer monsoon mean rainfall for SSP5-8.5 for all 32 models over the area displayed in Fig. 3. Relative change is calculated as the change in mean rainfall for the period 2070-2100 with respect to the period 1985-2015. The gap separates models with rainfall values for 1985-2015 within twice the standard deviation of the reanalysis mean as in Fig. 1 from those outside that range. Please notice the different scales in the two panels. The mean over all models is +24.3%.

dependence of interannual variability on global mean temperature for all available models (after removing the trend). As the global mean temperature change grows with stronger forcing, the positive trend in interannual variability becomes clearer.

## 4    Discussion

In this study, the long-term trend of the Indian summer monsoon and its variability have been analyzed based on the latest global coupled model simulations under the SSP scenarios. Our approach addresses the question whether the results from earlier studies can be confirmed or need to be adapted in their sign or magnitude.

By comparing the CMIP-6 projection results with the WFDE5 reanalysis data, we classified some models as probably more capable of simulating a realistic representation of the monsoon rainfall. The share of models that capture the reference rainfall

within twice the standard deviation has slightly increased in CMIP-6 (16 out of 32) in comparison to the precursor models in CMIP-5 (9 out of 20) (Menon et al., 2013). But it has to be noted that the validation period and the used reanalysis data differ between (Menon et al., 2013) and this study. The observation of quantitatively measurable improvement between CMIP-5 and CMIP-6 coincides with the results of Gusain et al. (2020). While all the models that were out of the two standard deviation range underestimated the mean in CMIP-5, thus revealing a very clear general tendency of underestimation, the 16 models





**Figure 7.** Difference in Indian summer monsoon mean rainfall (mm/day) for the period 2070-2100 under SSP5-8.5 in comparison to 1985-2015.

outside of the range in CMIP-6 partly underestimated (13 models) and party overestimated (3 models) the observed mean in
1985-2015. Modeling centers whose models underestimated the rainfall within two standard deviations in our study mostly
underestimated the rainfall already in CMIP-5. Some models with realistic patterns in CMIP-6 are updates from CMIP-5 that
already revealed a pattern relatively similar to reanalysis data, e.g. NorESM2-MM. As in CMIP-5, models with the tendency
to underestimate the rainfall in the evaluation period are mostly not capable either of capturing the spatial rainfall pattern in

CMIP-6. But there are also various models that improved their capacity in capturing the Indian monsoon, such as the models
from Centre National de Recherches Météorologiques (CNRM-CM6-1, CNRM-CM6-1-HR, CNRM-ESM2-1). This observed
inconsistency among models in improving their spatial representation of the Indian monsoon was already noted by Gusain





**Figure 7 (continued).**

et al. (2020). Besides, the capacity of capturing the rainfall pattern over the Western Ghats has improved what also coincides with the results of (Gusain et al., 2020).

The CMIP-6 models project a robust intensification of the Indian summer monsoon rainfall under climate change. All of the 32 available models exceed the envelope of baseline variability from 1850-2015 until 2100 under SSP5-8.5, while just 17 out of 20 exceeded the natural variability threshold under RCP-8.5 in a previous study based on CMIP-5 (Menon et al., 2013). Additionally, we calculated the average multi-model trend of projected change in mean rainfall by the end of the 21st century. As some modeling centers provide several models and some of them are based on overlapping model components, the

models cannot be regarded as independent from each other (See e.g., Knutti et al., 2017)). The results have to be interpreted against this background. The found average multi-model trend in CMIP-6 with an increase of +24.3% by 2100 seems stronger in comparison to CMIP-5 (Chaturvedi et al., 2012; Menon et al., 2013). As some modeling centers provide several models and





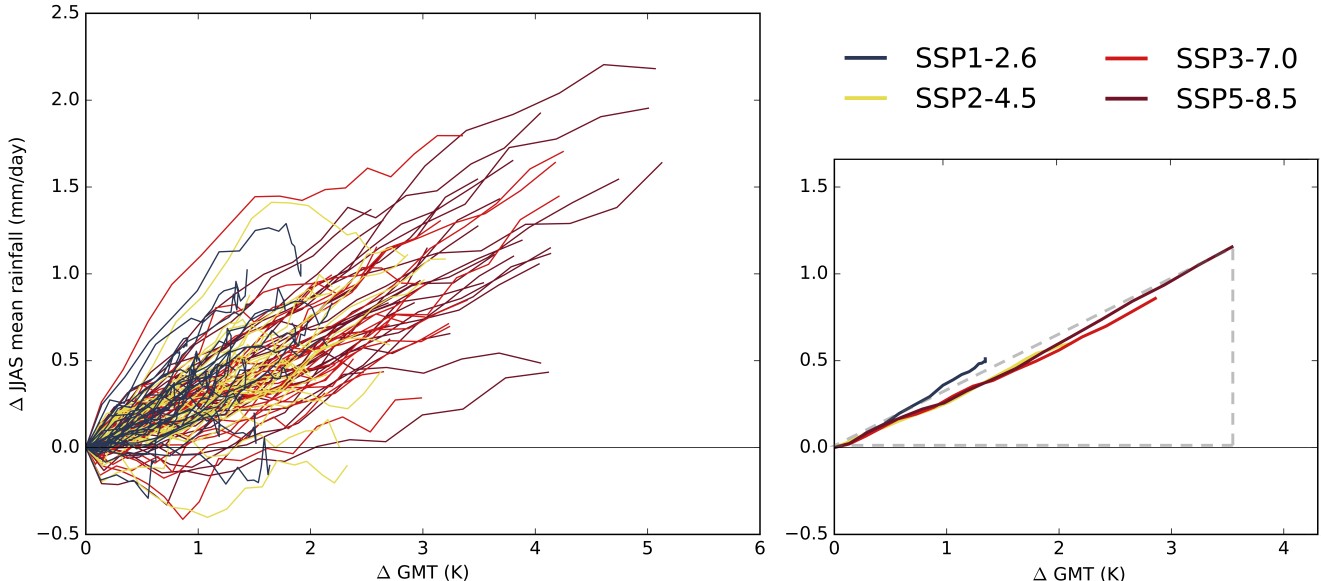

**Figure 8.** Change of Indian summer monsoon mean rainfall (mm/day) depending on change in global mean temperature (K) until the end of the 21st century relative to the period 1985-2015 for four scenarios. Underlying regions as in Fig. 3. Means are calculated over decadal periods starting in 2005 and overlapped by five years (2005-2014, 2010-2019, up to 2090-2099). Left: Each line represents a different model (one ensemble member per model). Right: Each line represents multi-model mean for one scenario. Model availability for global temperature in different scenarios can be seen in Table 1. Grey dashed lines indicate the slope (the hydrological sensitivity) for SSP5-8.5

some of them are based on overlapping model components, the models cannot be regarded as independent from each other. The results have to be interpreted against this background.

Chaturvedi et al. (2012) found an increase of 18.7% in RCP-8.5 by 2099 compared to the period 1961-1990 in CMIP-5 models. But because of the used time periods as well as the different study area of India without adjacent regions, this study is not directly comparable to ours. An intensification of the Indian monsoon rainfall also has been found in other studies using CMIP-5 (Lee and Wang, 2014; Mei et al., 2015; Sharmila et al., 2015; Varghese et al., 2020). There is a widespread agreement that a reason for the intensification of the South Asian monsoon rainfall is an increase in moisture convergence

(Singh et al., 2019). This enhanced thermodynamic effect dominates over the dynamic effect which refers to the decreasing monsoon circulation. In this way, the thermodynamic effect determines the positive sign of the change in monsoon rainfall whereas the dynamic effect would lead to a decrease in rainfall (Sooraj et al., 2015).

We found that the monsoon rainfall is linear dependent on the GMT. The projected increase in rainfall is 0.33 mm/day per degree of global warming. The agreement between models and the independence of the scenario is remarkable. The median

dependence of relative change in precipitation on GMT taking into account all models has increased from 3.2%/K in CMIP-5 to 5.3%/K in CMIP-6. Considering only the models with a more realistic representation of the monsoon, the increase is even more noticeably from 2.3%/K in CMIP-5 to 6.1%/K in CMIP-6. It also has to be mentioned that the range of projected

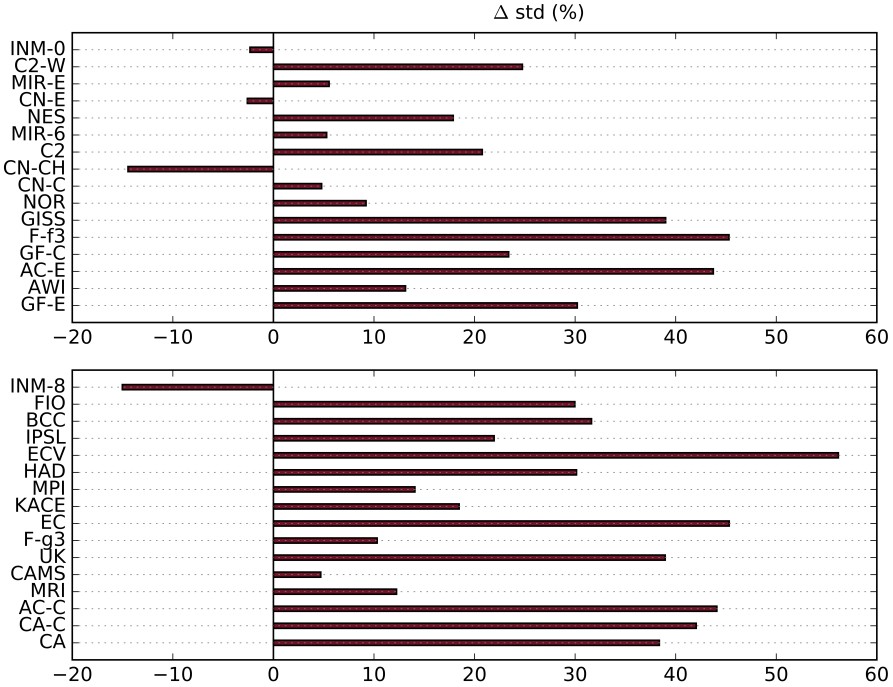

**Figure 9.** The percentage change of standard deviation between the second half of the 21st century to the standard deviation from 1900-1950 under SSP5-8.5. For the underlying area refer to Fig. 3. We used a Singular Spectrum Analysis algorithm (Golyandina and Zhigljavsky, 2013) to remove the nonlinear trend according to Fig. 4. The mean percentage change in this scenario is 21.3 %. The gap separates models as in Fig. 1 according to their capacity of capturing the monsoon rainfall in 1985-2015.

sensitivities has decreased remarkably from 1-19%/K in CMIP-5 to 2-13%/K in the latest generation of climate models, i.e. the uncertainty in hydrological sensitivity has decreased with the model updates. Similar tendencies have been found for the
equilibrium climate sensitivity in CMIP-6 Zelinka et al. (2020); Wyser et al. (2020). Which of the updated processes between CMIP-5 and CMIP-6 described by Gusain et al. (2020) dominate in causing the increased sensitivity of the monsoon to global warming needs further investigation.

The increase in rainfall is projected to contribute to the precipitation in the Himalaya region, the northeast Bay of Bengal and the northwest coast of India. These regions coincide to a large extent with the existing monsoon rainfall pattern, leading to a wet
regions get wetter pattern during June to September monsoon rainfall. The distribution of regions with projected increasing precipitation in CMIP-6 confirms the projection of previous studies using CMIP-5 models (Chaturvedi et al., 2012; Menon et al., 2013; Sharmila et al., 2015). Furthermore, the increasing pattern is shared by a larger percentage of available models in CMIP-6 compared to CMIP-5. But our projection of increased rainfall over the Western Ghats does not coincide with the study of Varghese et al. (2020) projecting a decrease in this region. By focusing on high resolution models with the best deep
convection scheme, their study reveals decreasing precipitation in the southwest coast of India, which is only captured by one





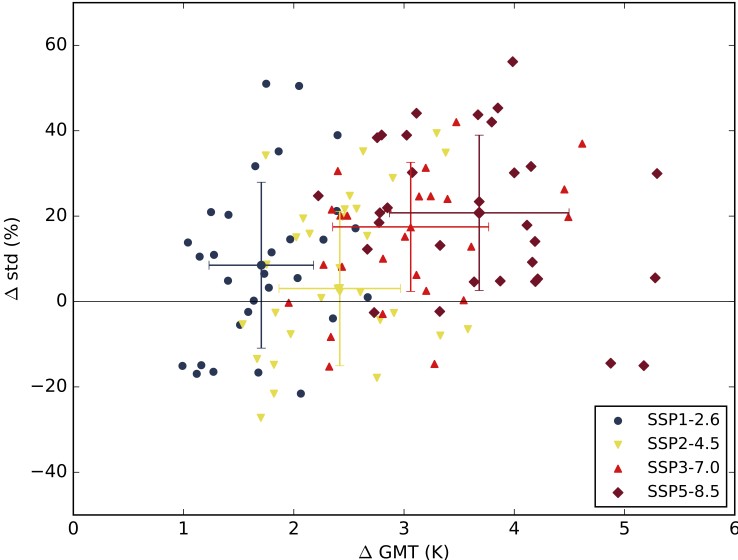

**Figure 10.** Scatterplot of percent change in standard deviation(%) and change in global mean temperature (K) between 2050-2100 and 1950-2000 for four scenarios. The symbols with error range represent the median plus/minus the standard deviation in each scenario. The underlying area can be seen in Fig. 3. The trend was removed before using a Singular Spectrum Analysis algorithm (Golyandina and Zhigljavsky, 2013). Availability of models for different scenarios can be seen in Table 1.

third of the CMIP-6 models in our study, including the CNRM-CM6-1-HR model. A finer resolution seems to be necessary to capture this trend which is not given for all CMIP-6 models.

From the 32 available models, 28 models project an increase in interannual variability. This result is not directly comparable to the study of Menon et al. (2013) since the removal of the trend in our study has a relevant influence on the results. Without the removal of the trend, i.e. following the method of Menon et al. (2013), all 32 models project an increase in interannual variability which shows that the signal has become clearer in comparison to the results in CMIP-5 models. The projected increase in interannual variability coincides with other studies (Kitoh et al., 1997; Jayasankar et al., 2015; Sharmila et al., 2015; Kitoh, 2017). A dominant role in shaping the interannual variability is taken by the El Niño Southern oscillation (ENSO) (Turner and Annamalai, 2012). As El-Niño events typically coincide with dry monsoon years and La-Niña years are often accompanied by strong monsoon rainfall (Kumar et al., 2006), changes in the emergence of these events have a relevant impact on the Indian summer monsoon. (Azad and Rajeevan, 2016) applied spectral analysis and found a shortening of the spectral periods of ENSO which might lead to a shift in the relationship of ENSO and monsoon rainfall.

In this study, we used 32 CMIP-6 models to analyse the Indian summer monsoon's response to climate change. In order to identify models with a good representation of the Indian monsoon, we compared the models' simulations in the past to WFDE5 reanalysis data. We found that there are 16 out of 32 models that are able to capture the monsoon rainfall within twice the standard deviation in the period 1985-2015. This is a slight increase compared to CMIP-5. The models outside that range in



CMIP-6 still have a tendency to underestimate the amount of precipitation in this period. This was already observed in CMIP-5 where all of the models out of the range underestimated the rainfall. In our analysis, we focused on the models with the more realistic representation of the Indian monsoon. We found that all models show an increase in mean summer monsoon rainfall under SSP5-8.5 and SSP3-7.0 by the end of the 21st century. An increase also was found in SSP2-4.5 and SSP1-2.6 by all models apart from two models in SSP2-4.5 and one model in SSP1-2.6. Under SSP5-8.5, the models exceed the envelope of the baselines variability on average in 2045 in SSP5-8.5. An multi-model mean increase of rainfall of 24,3% is projected under SSP5-8.5 and of +18,6% in SSP3-7.0, of +11,9% in SSP2-4.5 and of +9.7% in SSP1-2.6. The majority of models project that that the increase will contribute to the precipitation especially in the Himalaya region, the northeast of the Bay of Bengal and to the west coast of India. Besides, the simulation ensemble indicates a linear dependence of rainfall on global mean temperature independent of the SSP; the multi-model mean for JJAS projects an increase of 0.33mm/day and 5.3% per degree of global warming. Furthermore, under SSP5-8.5 a majority of 28 out of 32 models project an increase in interannual variability by the end of the 21st century after removing the trend with Singular Spectrum Analysis.

We have seen in this study that low resolution models did not capture the spatial pattern of the monsoon rainfall in historic periods well. Small scale topography and its atmosphere feedback influence the rainfall to a relevant extent. Thus, the ongoing effort to improve the resolution of the individual CMIP models should be continued. Since other rainfall features such as extremes and the variability of rainfall on a subseasonal scale are beyond the scope of this study, they need to be analyzed in further studies owing to their high relevance e.g. for high-risk flooding events.

The projected increase in summer monsoon rainfall in combination with the projected longterm increase in interannual variability will be accompanied by an increased number of extremely wet years and potentially more high rainfall events (Turner and Slingo, 2009; Sharmila et al., 2015). While crops need water especially in the initial growing period, high rainfall events during other growing states can harm the plants Revadekar and Preethi (2012). Thus, the projected development might have serious consequences for the agriculture in India and neighbouring regions. Since the change differs from the decreasing tendency in the second half of the 20th century, the development of adaptation strategies for the 21st century is required.

**Appendix A:  Appendix A: Change in Indian mean summer monsoon rainfall**

**Appendix D:  Appendix B: Change in Indian summer monsoon rainfall interannual variability**



**Table 1.** Overview of data availability for the 32 models used in the study (precipitation/temperature). Only those models are selected for which data for historic period and SSP5-8.5 was available at the time of the study.

| Modeling Center (Group) | Model | SSP1-2.6 | SSP2-4.5 | SSP3-7.0 | SSP5-8.5 |
|---|---|---|---|---|---|
| Alfred Wegener Institute (AWI) | AWI-CM-1-1-MR | Y/N | Y/N | Y/N | Y/N |
| Beijing Climate Center, China Meteorological Administration (BCC) | BCC-CSM2-MR | Y/Y | Y/Y | Y/Y | Y/Y |
| Chinese Academy of Meteorological Sciences (CAMS) | CAMS-CSM1-0 | Y/Y | Y/Y | Y/Y | Y/Y |
| LASG, Institute of Atmospheric Physics, Chinese Academy of Sciences (CAS) | FGOALS-f3-L | Y/Y | Y/Y | Y/Y | Y/Y |
|  | FGOALS-g3 | Y/Y | Y/Y | Y/Y | Y/Y |
| Canadian Centre for Climate Modelling and Analysis (CCCma) | CanESM5 | Y/Y | Y/Y | Y/Y | Y/Y |
|  | CanESM5-CanOE | Y/Y | Y/Y | Y/Y | Y/Y |
| Centre National de Recherches Météorologiques/ Centre Européen de Recherche et Formation Avancées en Calcus Scientifique (CNRM-CERFACS) | CNRM-CM6-1 | Y/Y | Y/Y | Y/Y | Y/Y |
|  | CNRM-CM6-1-HR | Y/Y | Y/Y | Y/Y | Y/Y |
|  | CNRM-ESM2-1 | Y/Y | Y/Y | Y/Y | Y/Y |
| Commonwealth Scientific and Industrial Research Organisation (CSIRO) | ACCESS-ESM1-5 | Y/Y | Y/Y | Y/Y | Y/Y |
| Commonwealth Scientific and Industrial Research Organisation, ARC Centre of Excellence for Climate System Science (CSIRO-ARCCSS) | ACCESS-CM2 | Y/Y | Y/Y | Y/Y | Y/Y |
| EC-Earth-Consortium | EC-Earth3 | Y/Y | Y/Y | Y/Y | Y/Y |
|  | EC-Earth3-Veg | Y/Y | Y/Y | Y/Y | Y/Y |
| First Institution of Oceanography (FIO-QLNM) | FIO-ESM-2-0 | Y/Y | Y/Y | N/N | Y/Y |
| Institute of Numerical Mathematics (INM) | INM-CM4-8 | Y/Y | Y/Y | Y/Y | Y/Y |
|  | INM-CM5-0 | Y/Y | Y/Y | Y/Y | Y/Y |
| Institut Pierre Simon Laplace (IPSL) | IPSL-CM6A-LR | Y/Y | Y/Y | Y/Y | Y/Y0 |
| Japan Agency for Marine-Earth Science and Technology/ Atmosphere and Ocean Research Institute, University of Tokyo (MIROC) | MIROC6 | Y/Y | Y/Y | Y/Y | Y/Y |
|  | MIROC-ES2l | Y/Y | Y/Y | Y/Y | Y/Y |
| Met Office Hadley Centre (MOHC) | HadGEM3-GC31-LL | Y/Y | Y/Y | N/N | Y/Y |
|  | UKESM1-0-LL | Y/Y | Y/Y | Y/Y | Y/Y |
| Max Planck Institute for Meteorology (MPI-M) | MPI-ESM1-2-LR | Y/Y | Y/Y | Y/Y | Y/Y |
| Meteorological Reserarch Institute (MRI) | MRI-ESM2-0 | Y/Y | Y/Y | Y/Y | Y/Y |
| NASA Goddard Institute for Space Studies (NASA-GISS) | GISS-E2-1-G | Y/Y | Y/Y | Y/Y | Y/Y |




| Modeling Center (Group) | Model | SSP1-2.6 | SSP2-4.5 | SSP3-7.0 | SSP5-8.5 |
|---|---|---|---|---|---|
| National Center for Atmospheric Research (NCAR) | CESM2 | Y/Y | Y/Y | N/N | Y/Y |
| | CESM2-WACCM | Y/Y | Y/Y | Y/Y | Y/Y |
| Norwegian Climate Center (NCC) | NorESM2-MM | Y/Y | Y/Y | Y/Y | Y/Y |
| National Institute of Meteorological Sciences-Korea Met. Administration (NIMS-KMA) | KACE-1-0-G | Y/Y | Y/Y | Y/Y | Y/Y |
| NOAA Geophysical Fluid Dynamics Laboratory (NOAA-GFDL) | GFDL-CM4 | N/N | Y/Y | N/N | Y/Y |
| | GFDL-ESM4 | Y/Y | Y/Y | Y/Y | Y/Y |
| Nanjing University of Information Science and Technology (NUIST) | NESM3 | Y/Y | Y/Y | N/N | Y/Y |
| | Number of models per scenario | 31/30 | 32/31 | 27/26 | 32/31 |

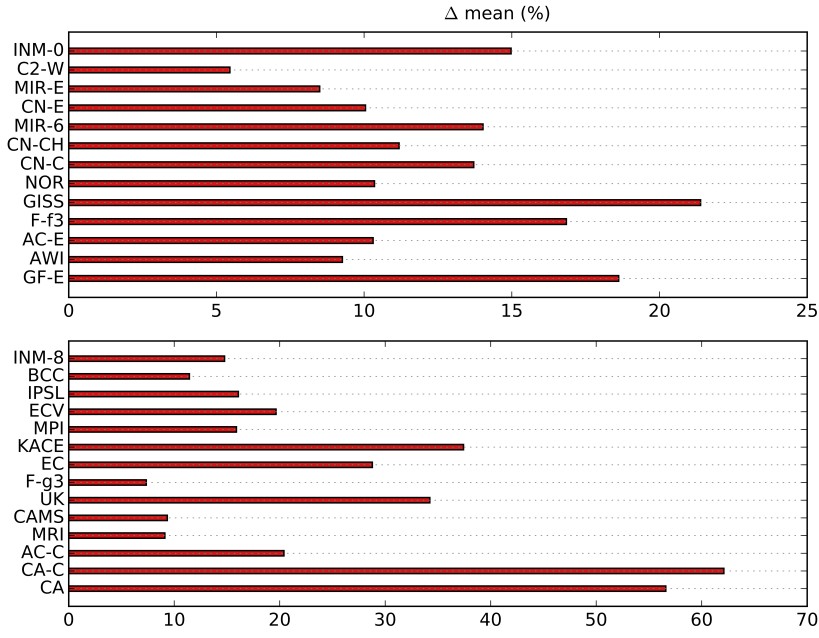

**Figure A1.** As in Fig. but for SSP3-7.0. The mean over all models is +18.6%. Please notice the different scales in the lower pannel.





**Table 2.** Overview of short names used in this study and resolution in which the 32 models were run.

| Model | Short Name | Atmosphere [km] | Land [km] | Ocean [km] |
|---|---|---|---|---|
| AWI-CM-1-1-MR | AWI | 100 | 100 | 25 |
| BCC-CSM2-MR | BCC | 100 | 100 | 50 |
| CAMS-CSM1-0 | CAMS | 100 | 100 | 100 |
| FGOALS-f3-L | F-f3 | 100 | 100 | 100 |
| FGOALS-g3 | F-g3 | 250 | 250 | 100 |
| CanESM5 | CA | 500 | 500 | 100 |
| CanESM5-CanOE | CA-C | 500 | 500 | 100 |
| CNRM-CM6-1 | CN-C | 250 | 250 | 100 |
| CNRM-CM6-1-HR | CN-CH | 100 | 100 | 25 |
| CNRM-ESM2-1 | CN-E | 250 | 250 | 100 |
| ACCESS-ESM1-5 | AC-E | 250 | 250 | 100 |
| ACCESS-CM2 | AC-C | 250 | 250 | 100 |
| EC-Earth3 | EC | 100 | 100 | 100 |
| EC-Earth3-Veg | ECV | 100 | 100 | 100 |
| FIO-ESM-2-0 | FIO | 100 | 100 | 100 |
| INM-CM4-8 | INM-8 | 100 | 100 | 100 |
| INM-CM5-0 | INM-0 | 100 | 100 | 50 |
| IPSL-CM6A-LR | IPSL | 250 | 250 | 100 |
| MIROC6 | MIR6 | 250 | 250 | 100 |
| MIROC-ES2l | MIR-E | 500 | 500 | 100 |
| HadGEM3-GC31-LL | HAD | 250 | 250 | 100 |
| UKESM1-0-LL | UK | 250 | 250 | 100 |
| MPI-ESM1-2-LR | MPI | 250 | 250 | 250 |
| MRI-ESM2-0 | MRI | 100 | 100 | 100 |
| GISS-E2-1-G | GISS | 250 | 250 | 100 |
| CESM2 | C2 | 100 | 100 | 100 |
| CESM2-WACCM | C2-W | 100 | 100 | 100 |
| NorESM2-MM | NOR | 100 | 100 | 100 |
| KACE-1-0-G | KACE | 250 | 250 | 100 |
| GFDL-CM4 | GF-C | 100 | 100 | 25 |
| GFDL-ESM4 | GF-E | 100 | 100 | 50 |
| NESM3 | NES | 250 | 2.5 | 100 |



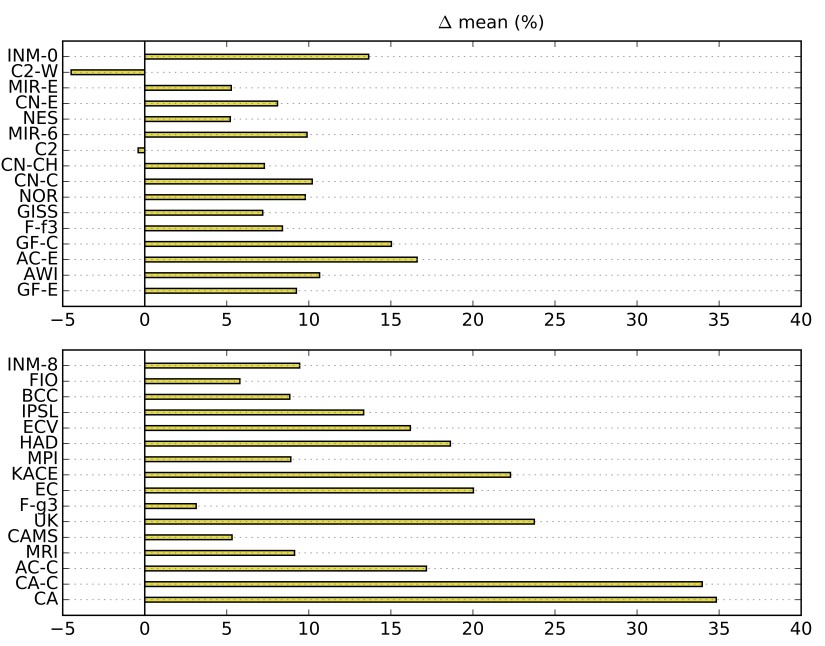

**Figure B1.** As in Fig. but for SSP2-4.5. The mean over all models is +11.9%

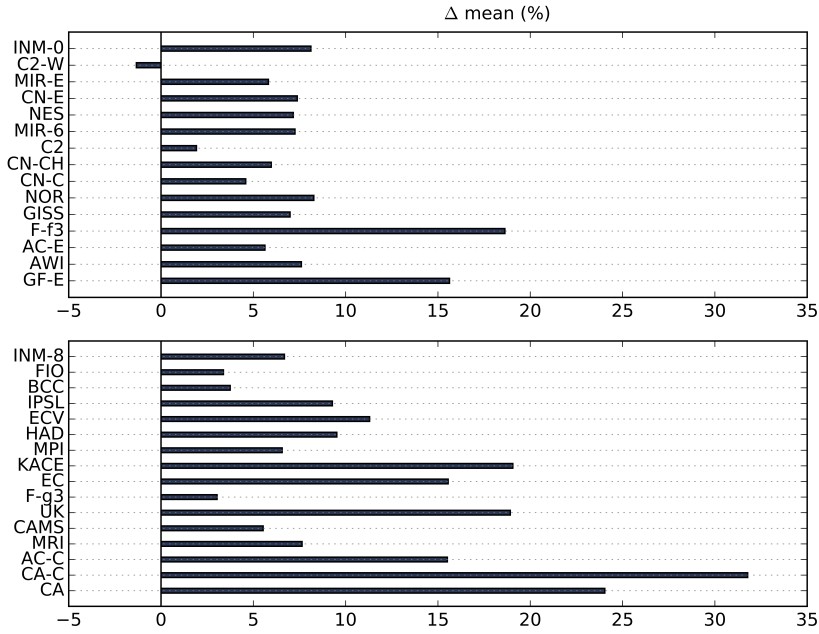

**Figure C1.** As in Fig. but for SSP1-2.6. The mean over all models is +9.7%



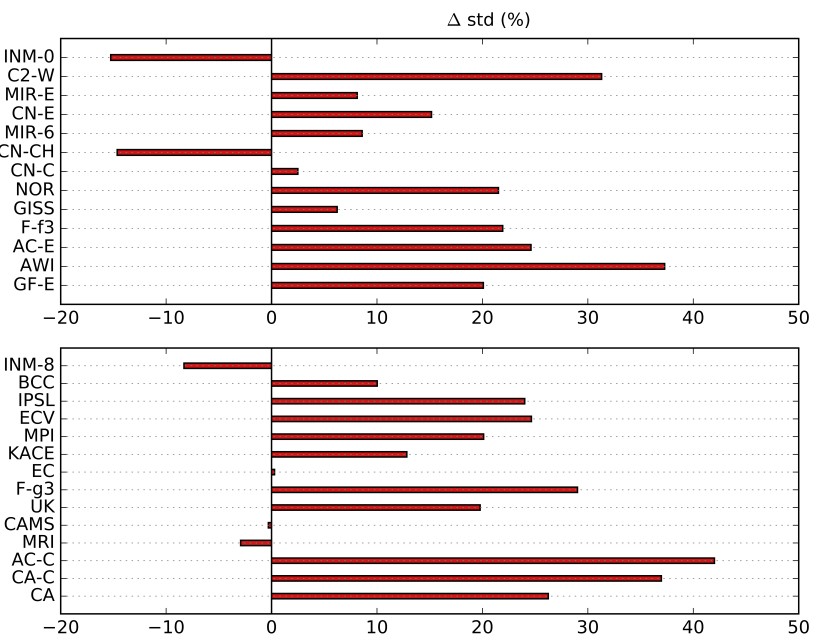

**Figure E1.** As in Fig. but for SSP3-7.0.

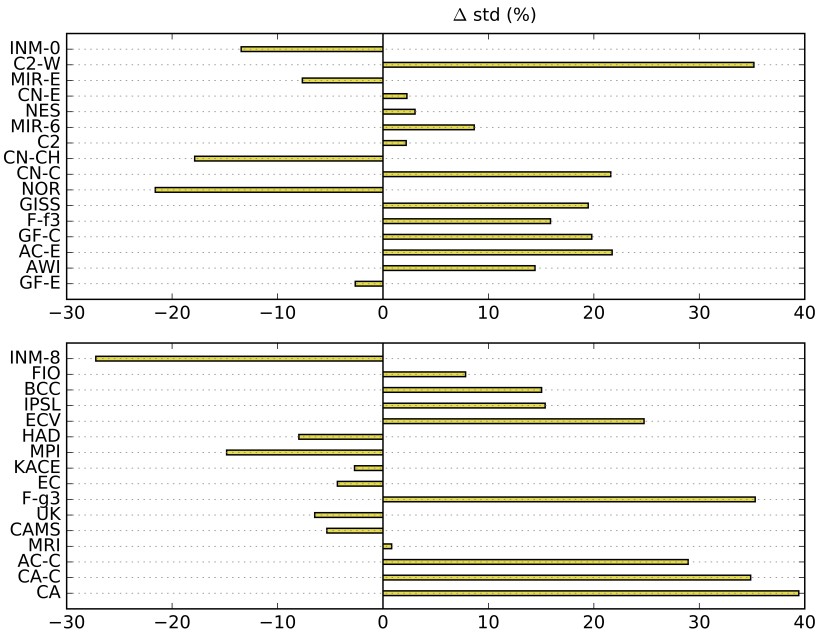

**Figure F1.** As in Fig. but for SSP2-4.5.





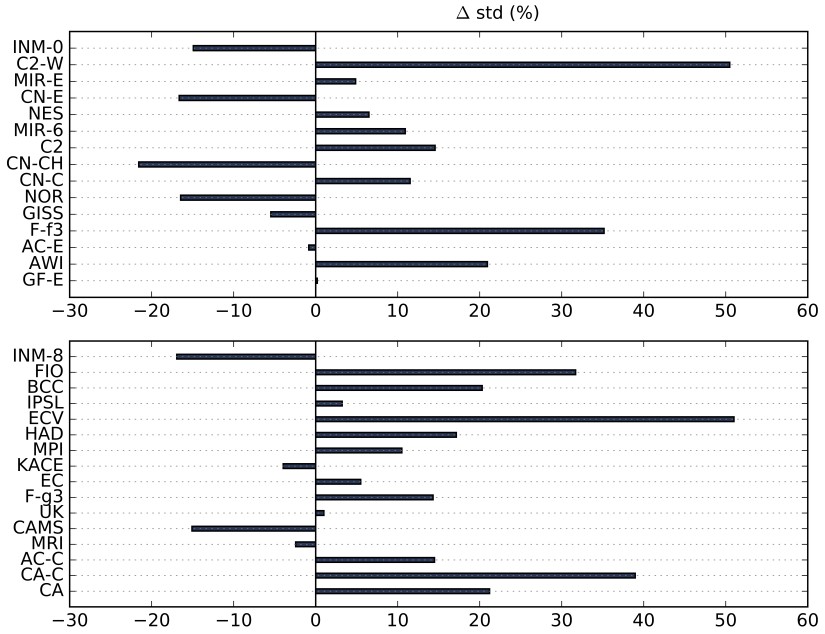

**Figure G1.** As in Fig. but for SSP1-2.6.

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
