# Peer review of "Robust increase of Indian monsoon rainfall and its variability under future warming in CMIP-6 models"

_Earth System Dynamics, 2020_

## Referee Comment (RC1) · Anonymous Referee #1 · 25 Nov 2020

This manuscript discusses expected future changes in Indian Monsoon precipitation amounts and variability. To this end, the authors use 32 CMIP-6 model projections under different shared socioeconomic pathways: SSP1-2.6, SSP2-4.5, SSP3-7.0 and SSP5-8.5. The main conclusions are that the Indian Monsoon will experience an increase in precipitation and its interannual variability will become larger. The methods are standard in this type of analysis, the results are properly discussed and well supported by easy to follow figures. The final discussion does a good summary of the findings. While the results would stand on their own, they seem to contradict previous studies summarized in the introduction. This discrepancy needs to be resolved before the manuscript is accepted for publication. There are also several minor comments

that will need to be corrected.

1. The main conclusions in this study are that regardless of the shared socioeconomic pathway considered, most models show an increase in precipitation in the Indian Monsoon. Even more, the simulation ensemble indicates a linear dependence of rainfall on global mean temperature with high agreement between the models and independent of the SSP. These results contradict previous observational studies (see introduction) where it was shown that there was a decreasing trend of precipitation in the second half of the 20th century. Those same studies suggest possible physically-based reasons for the decreasing trend. If there was a decrease in the 20th century precipitation, it would contradict the assumption of a linear relation between rainfall and temperature. What changed to start having the positive trend discussed in this manuscript? Can you offer a justification that explains the perceived differences between the 20th and 21st centuries?

2. The figures show the projections for each model. The ensembles should be included as well.

3. A thorough review is necessary. Note, for example, the incomplete sentence in line 111, or the repeated text around lines 239-252 and 242-244.

4. All Figure Captions in the Appendices are incomplete: They all state: "According to Fig. (missing number)"

Minor comments: Line 22: what is a retractable effect? Line 101 onle –> nly Lines 106, 249 It is "moisture flux convergence". Moisture (a scalar) does not converge. Moisture flux (a vector) does. There may be other instances with this mistake Line 108: Walker is uppercase Line 250: Moisture (flux) convergence is not a thermodynamic effect. It involves moisture, but it is a dynamic effect as well. It is mostly driven by the convergence of winds.

---

## Author Comment (AC1) · 4 Dec 2020

We thank the reviewer for their constructive criticism, and time spent to analyze this manuscript. The responses, and explanations related to their comments are listed below (in green).

This manuscript discusses expected future changes in Indian Monsoon precipitation amounts and variability. To this end, the authors use 32 CMIP-6 model projections under different shared socioeconomic pathways: SSP1-2.6, SSP2-4.5, SSP3-7.0 and SSP5-8.5. The main conclusions are that the Indian Monsoon will experience an increase in precipitation and its interannual variability will become larger. The methods are standard in this type of analysis, the results are properly discussed and well supported by easy to follow figures. The final discussion does a good summary of the findings. While the results would stand on their own, they seem to contradict previous studies summarized in the introduction. This discrepancy needs to be resolved before the manuscript is accepted for publication. There are also several minor comments that will need to be corrected.

1. The main conclusions in this study are that regardless of the shared socioeconomic pathway considered, most models show an increase in precipitation in the Indian Monsoon. Even more, the simulation ensemble indicates a linear dependence of rainfall on global mean temperature with high agreement between the models and independent of the SSP. These results contradict previous observational studies (see introduction) where it was shown that there was a decreasing trend of precipitation in the second half of the 20th century.

Response: Please note that the observed decreasing trend in the second half of the 20th century described in the introduction can also be found in the ensemble mean of the models as displayed in figure 05. Thus, observations are not opposing the model results.

Those same studies suggest possible physically-based reasons for the decreasing trend. If there was a decrease in the 20th century precipitation, it would contradict the assumption of a linear relation between rainfall and temperature.

Response: This is an important aspect and we would like to thank the reviewer for pointing this out. We will add a discussion of the matter in a revised manuscript. The reason why we believe that there is no real contradiction in the physics (but obviously in the manuscript which we will resolve in the revised manuscript) is the following: We analyzed the dependence of increase in annual monsoon rainfall on global mean temperature in the 21st century relative to 1985-2015 which is a period when the GHG warming and associated warming over land is expected to dominate the Indian summer monsoon dynamics. Thus, the found linear relationship is valid when GHG warming is the dominant factor. We added an explaining comment for clarifying more precisely when the found relationship is valid ("The simulation ensemble indicates a linear dependence of rainfall on global mean temperature with high agreement between the models and independent of the SSP if global warming is the dominant forcing of the monsoon dynamics as it is in the 21st century"). Besides, we added and explaining paragraph in the discussion: "The simulation ensemble indicates a linear dependence of rainfall on global mean temperature with high agreement between the models and independent of the SSP. This is not in contradiction with the observed decline in monsoon rainfall during the second half of the 20th century: While between the 1950s and 70s, approximately, high aerosol loadings led to subdued warming and a weakened land-sea thermal gradient, greenhouse gas-induced warming has dominated since then and is the dominant forcing in the 21st century projections." Nevertheless, we assume that this relationship would also be found if a period of the 20th century was to be included in the calculation since the increase in the 21st century is much stronger than the decrease in the second half of the 20th century (as can been seen in figure 04 & 05). Besides, please note that there was actually a stagnation of global temperature increase in 1950-1970.

What changed to start having the positive trend discussed in this manuscript?

Response: The land warming due to GHG emissions becomes dominant over the former forcing factors (e.g. dampened land warming due to aerosol emissions). This leads to an increased temperature gradient in the lower troposphere resulting in more summer monsoon rainfall. We explain this in line 76-82. Nevertheless, we do agree that the emphasize in the introduction on the forcing factors leading to a decrease of rainfall since the 1950s might be misleading and thus we changed the focus concentrating more on the factors relevant for the increasing temperature gradient analyzed in our study.

Can you offer a justification that explains the perceived differences between the 20th and 21st centuries?

**Response:** In the 20th century the forcing is dominating the Indian monsoon dynamics differ from the ones in the 21st century as explained above.

2. The figures show the projections for each model. The ensembles should be included as well.

**Response:** Thanks for this comment, we think displaying the ensemble mean is a valuable addition for figure 06, figure 09 as well as the figures in the Supplementary Information (vertical lines to mark the ensemble mean). Regarding figure 04, please note that figure 05 is already providing an ensemble mean for this figure. In order to make this clearer for the reader, we added: "For the multi-model mean under SSP5-8.5 and other scenarios refer to Fig. 5" in the figure caption of Fig. 4.".

3. A thorough review is necessary. Note, for example, the incomplete sentence in line 111, or the repeated text around lines 239-252 and 242-244.

**Response:** Lines 111, 239-252 and 242-244 were corrected. A thorough review has been done.

4. All Figure Captions in the Appendices are incomplete: They all state: "According to Fig. (missing number)"

**Response:** We added missing numbers in the figure captions in the Supplementary Information.

**Minor comments:**

Line 22: what is a retractable effect?

**Response:** We clarified the meaning in line 22.

Line 101 onle –> nly

**Response:** We corrected the typing error in line 101.

Lines 106, 249 It is "moisture flux convergence". Moisture (a scalar) does not converge. Moisture flux (a vector) does. There may be other instances with this mistake

**Response:** We corrected line 106 and 249 and checked if there are other instances with this mistake.

Line 108: Walker is uppercase

**Response:** Corrected.

Line 250: Moisture (flux) convergence is not a thermodynamic effect. It involves moisture, but it is a dynamic effect as well. It is mostly driven by the convergence of winds.

**Response:** We refer to the wording of Sooraj et al. 2015 which can be found as well e.g. in Mei et al. 2015 (doi: 10.1175/JCLI-D-14-00355.1), Cherchi et al. 2011 (doi:10.1007/s00382-010-0801-7) and Endo & Kitoh 2014 (doi: 10.1002/2013GL059158). The term seemingly refers to the origin of the dynamics which in this case is based on the thermodynamic gradient.

---

## Referee Comment (RC2) · Anonymous Referee #2 · 22 Dec 2020

Review of "Robust increase of Indian monsoon rainfall and its variability under future warming in CMIP-6 models" by Anja Katzenberger, Jacob Schewe, Julia Pongratz, and Anders Levermann

This study analyses change in Indian summer monsoon in a set of models from different CMIP6 scenarios. Authors found a long-term increase of Indian summer monsoon precipitation and an increased of its interannual variability. The paper is quite a description of Indian summer monsoon model results from newest generation of CMIP. The paper confirms the increased long term trend in Indian monsoon precipitation already found by previous CMIP models, as well its interannual variability (with some dif-

ferences). Authors did not investigate what drives the large difference found at regional scale on monsoon response in the different models, which I think it is quite interesting. They just mentioned that the resolution matters (still). Overall, the paper addresses the questions within the ESD scope. It shows some new results based on new data available and conclusions are reached. Only Introduction needs substantial revision because it is a bit chaotic. I recommend to publish the paper after major revision.

Ln 35-39: "Multi-millennial paleorecords indicate strong changes both in the Indian and East Asian summer monsoon (Wang et al., 2005b, a, 2008; Zhang et al., 2008; Li et al., 2017; Wang et al., 2017; Zhang et al., 2019; Ming et al.; Wang et al., 2020). While it is speculated (Schewe et al., 2012; Herzschuh et al., 2014; Wang et al., 2020), that there might be abrupt monsoon changes due to a moisture-advection feedback at play (Levermann et al., 2009), these are generally associated with either aerosol forcing or changes in the sea surface temperatures of the surrounding ocean waters."

This sentence is quite generic. What multi-millennial paleorecords are you referring to here? Are this changes related to orbital parameters during the Holocene? And in particular this sentence " . . . that there might be abrupt monsoon changes due to a moisture-advection feedback at play (Levermann et al., 2009), these are generally associated with either aerosol forcing or changes in the sea surface temperatures of the surrounding ocean waters." is totally misleading. Aerosol forcing on multi-millennial time scales? No-way. I warmly suggest to rephrase here. Do not mix too much. If you really want to refer to both past and future Indian monsoon changes, you might find useful this paper for both contents and recent literature overview.

D'Agostino, R., Bader, J., Bordoni, S., Ferreira, D., & Jungclaus, J. (2019). Northern Hemisphere Monsoon Response to Mid-Holocene Orbital Forcing and Greenhouse Gas-Induced Global Warming.Geophysical Research Letters,46(3),1591-1601.

Ln 39-40: "Under future warming an overall strengthening of the monsoon rainfall is expected due to enhanced atmospheric moisture bearing capacity." Please add a reference here.

Ln 42-43: "The resulting decrease in the land-sea thermal gradient over South Asia and the consequently subdued Hadley circulation have lead to a reduction of the rainfall amount during the summer period since the 1950s (Roxy et al., 2015)." Try to expand a bit here.

Ln 45-82: These paragraphs are totally confusing. You are trying to summarise in a chaotic way three decades of studies about Hadley Circulation and monsoons, meridional and land/sea temperature contrasts influence on monsoon dynamics, oceanic warming, ENSO, aerosols, vegetation, energy budget... too much, not effective and not focussed. I strongly suggest to rewrite the section trying to put things in a clear way. You can list the different monsoon response sorting by the type of forcing for example. E.g. GHG vs aerosols or envisaging monsoon response in terms of moist static energy budget and energy framework.

Refer to:

Allan, R., Barlow, M., Byrne, M. P., Cherchi, A., Douville, H., Fowler, H. J., ... & Wilcox, L. (2020). Advances in understanding large-scale responses of the water cycle to climate change.Annals of the New York Academy of Sciences

Boos, W. R., & Korty, R. L. (2016). Regional energy budget control of the intertropical convergence zone and application to mid-Holocene rainfall.Nature Geoscience,9(12),892-897.

D'Agostino, R., Brown, J. R., Moise, A., Nguyen, H., Dias, P. L. S., & Jungclaus, J. (2020). Contrasting Southern Hemisphere Monsoon Response: MidHolocene Orbital Forcing versus Future Greenhouse Gas-Induced Global Warming.Journal of Climate,33(22),9595-9613.

Jalihal, C., Srinivasan, J., & Chakraborty, A. (2019). Modulation of Indian monsoon by water vapor and cloud feedback over the past 22,000 years.Nature

communications,10(1),1-8.

Seth, A., Giannini, A., Rojas, M., Rauscher, S. A., Bordoni, S., Singh, D., & Camargo, S. J. (2019). Monsoon responses to climate changes-connecting past, present and future.Current Climate Change Reports,5(2),63-79.

Ln 101: "...onle..." Typo. Ln 104-105: "Also under SSP5-8.5, the amount of rainfall over India is projected to increase by 18.7% by the end of the 21st century compared to 1961-1999 (Chaturvedi et al., 2012)." I thought that SSP5-8.5 is the newest experiment under CMIP6. How can be the ref so old? Maybe a typo?

Ln 107-108: about the thermodynamics vs dynamics add as ref D'Agostino et al., 2019 and 2020.

Ln 111: "The uncertain role of ..." Missing something here.

Ln 126: "67.5°0'0"E - 98°0'0"E and latitude 6°0'0"N-36°0'0"N". I do not think you need coordinates in minutes and seconds here.

Ln 250-253: refer to aforementioned studies about thermodynamics vs dynamics. Ln 253: linear -> linearly

Ln 213: Discussion... and Conclusions?

Ln 283: "In this study, we used 32 CMIP-6 models to analyse the Indian summer monsoon's response to climate change." I would not repeat "in this study...".

---

## Author Comment (AC2) · 15 Jan 2021

We thank the reviewer for their time spent to analyze the manuscript and the constructive criticism. Particularly, we thank the reviewer for sharing his/her impression of the confusing paragraphs in the introduction which motivated us for a clearer structuring of contents in the introduction in the revised manuscript.

**This study analyses change in Indian summer monsoon in a set of models from different CMIP6 scenarios. Authors found a long-term increase of Indian summer monsoon precipitation and an increased of its interannual variability. The paper is quite a description of Indian summer monsoon model results from newest generation of CMIP. The paper confirms the increased long term trend in Indian monsoon precipitation already found by previous CMIP models, as well its interannual variability (with some differences). Authors did not investigate what drives the large difference found at regional scale on monsoon response in the different models, which I think it is quite interesting. They just mentioned that the resolution matters (still). Overall, the paper addresses the questions within the ESD scope. It shows some new results based on new data available and conclusions are reached. Only Introduction needs substantial revision because it is a bit chaotic. I recommend to publish the paper after major revision.**

Ln 35-39: "Multi-millennial paleorecords indicate strong changes both in the Indian and East Asian summer monsoon (Wang et al., 2005b, a, 2008; Zhang et al., 2008; Li et al., 2017; Wang et al., 2017; Zhang et al., 2019; Ming et al.; Wang et al., 2020). While it is speculated (Schewe et al., 2012; Herzschuh et al., 2014; Wang et al., 2020), that there might be abrupt monsoon changes due to a moisture-advection feedback at play (Levermann et al., 2009), these are generally associated with either aerosol forcing or changes in the sea surface temperatures of the surrounding ocean waters."

This sentence is quite generic. What multi-millennial paleorecords are you referring to here? Are this changes related to orbital parameters during the Holocene?

**Response:** We agree that additional information might give the reader better understanding of the paleorecords and the underlying forcing which is why we added information to clarify which paleo-records we are referring to:

Multi-millennial paleorecords indicate strong changes both in the Indian and East Asian summer monsoon. These paleoclimatic changes have been revealed by e.g. oxygen isotope analysis from different caves in Asia for the past thousands of years (Wang et al., 2008; Zhang et al., 2008, 2019, Wang et al., 2005b), by analysing marine sediment records for the Neogene and Quaternary (Wang et al., 2005a), and other methods (Li et al., 2017; Wang et al., 2017, Ming et al., 2020, Wang et al. 2020). Most studies link the paleoclimatic changes of monsoon rainfall predominantly to solar insolation variations on the northern hemisphere affecting the ITCZ position due to orbital forcing changes (Wang et al., 2005a, b, 2008; Zhang et al., 2008, 2019; Ming et al., 2020).

And in particular this sentence " ... that there might be abrupt monsoon changes due to a moisture-advection feedback at play (Levermann et al., 2009), these are generally associated with either aerosol forcing or changes in the sea surface temperatures of the surrounding ocean waters." is totally misleading. Aerosol forcing on multi-millennial time scales? No-way. I warmly suggest to rephrase here. Do not mix too much. If you really want to refer to both past and future Indian monsoon changes, you might find useful this paper for both contents and recent literature overview.

D'Agostino, R., Bader, J., Bordoni, S., Ferreira, D., & Jungclaus, J. (2019). Northern Hemisphere Monsoon Response to Mid-Holocene Orbital Forcing and Greenhouse Gas-Induced Global Warming.Geophysical Research Letters,46(3),1591-1601.

**Response:** We thank the referee for raising the point that it might not be clear where we are referring to potential future and where to past changes. Thus, in the revised manuscript we focused in this paragraph on what was found in paleorecords excluding potential future changes. Besides, we think that the paper of D'Agostino et al. (2019) you proposed, contains information interesting and relevant for this publication but since we decided not to include future Indian monsoon changes in this paragraph, we included it later in the revised manuscript.

Especially to explain abrupt non-linear monsoon transitions as observed in the Holocene in the Tibetan Plateau, gradual insolation changes are not sufficient and thus, internal feedback mechanisms seem to be at play (Schewe et al., 2012; Herzschuh et al., 2014; Boos and Korty, 2016; Wang et al., 2020). The moisture-advection feedback (Levermann et al., 2009) might be such an internal mechanism that is able to provoke abrupt transitions and might be responsible for the abrupt Tibetan Plateau transitions in the Holocene (Herzschuh et al., 2014). Other amplifying effects might have occurred due to a water vapour and cloud feedback (Jalihal et al., 2019).

Ln 39-40: "Under future warming an overall strengthening of the monsoon rainfall is expected due to enhanced atmospheric moisture bearing capacity." Please add a reference here.

**Response:** This sentence has been removed in the context of restructuring the introduction. For reference, refer to:

Turner, A., G., Annamalai, H. (2012): Climate change and the South Asian summer monsoon. In: Nature Climate Change 2, 587-595.

Ln 42-43: "The resulting decrease in the land-sea thermal gradient over South Asia and the consequently subdued Hadley circulation have lead to a reduction of the rainfall amount during the summer period since the 1950s (Roxy et al., 2015)." Try to expand a bit here.

**Response:** We added further explanation in the revised manuscript:

The resulting decrease in the land-sea thermal gradient over South Asia opposes the pressure gradient driving the Hadley circulation and consequently subdues the Hadley circulation. Since the Hadley system is responsible for transporting the rainfall to the subcontinent, this is accompanied by a reduction of the rainfall amount during the summer period as observed since the 1950s (Roxy et al. 2015).

Ln 45-82: These paragraphs are totally confusing. You are trying to summarise in a chaotic way three decades of studies about Hadley Circulation and monsoons, meridional and land/sea temperature contrasts influence on monsoon dynamics, oceanic warming, ENSO, aerosols, vegetation, energy budget... too much, not effective and not focussed. I strongly suggest to rewrite the section trying to put things in a clear way. You can list the different monsoon response sorting by the type of forcing for example. E.g. GHG vs aerosols or envisaging monsoon response in terms of moist static energy budget and energy framework.

Refer to:

Allan, R., Barlow, M., Byrne, M. P., Cherchi, A., Douville, H., Fowler, H. J., ... & Wilcox, L. (2020). Advances in understanding large-scale responses of the water cycle to climate change. Annals of the New York Academy of Sciences

Boos, W. R., & Korty, R. L. (2016). Regional energy budget control of the intertropical convergence zone and application to mid-Holocene rainfall. Nature Geoscience,9(12),892-897.

D'Agostino, R., Brown, J. R., Moise, A., Nguyen, H., Dias, P. L. S., & Jungclaus, J. (2020). Contrasting Southern Hemisphere Monsoon Response: MidHolocene Orbital Forcing versus Future Greenhouse Gas-Induced Global Warming. Journal of Climate,33(22),9595-9613.

Jalihal, C., Srinivasan, J., & Chakraborty, A. (2019). Modulation of Indian monsoon by water vapor and cloud feedback over the past 22,000 years. Nature communications,10(1),1-8.

Seth, A., Giannini, A., Rojas, M., Rauscher, S. A., Bordoni, S., Singh, D., & Camargo, S. J. (2019). Monsoon responses to climate changes-connecting past, present and future. Current Climate Change Reports,5(2),63-79.

**Response:** We thank the reviewer for sharing his impression which motivated us to restructure the central paragraphs in the Introduction. In this context, we decided to emphasize the competing effects of GHG and aerosol forcing as proposed from the reviewer and structured the paragraphs according to different forcings present in multi-millennial paleorecords and observations since the 1950s.

Besides, we thank the reviewer for the recommended additional information, e.g. Seth et al. (2019) provides a valuable overview close to the content of our paragraph which is why we included this reference, but also Boos et al. (2016), Jalihal et al. (2019) and Allan et al. (2020). Since the introduction is becoming pretty long, we tried to keep it as short as possible. Since D'Agostino et al. (2020) analyses the monsoon responses on the Southern Hemisphere, we think that exceeds the scope of the introduction and might even create more confusion through opening a new topic which is why we decided not to include it in the revised manuscript.

Ln 101: ". . .onle. . ." Typo.

**Response:** Corrected.

Ln 104-105: "Also under SSP5-8.5, the amount of rainfall over India is projected to increase by 18.7% by the end of the 21st century compared to 1961-1999 (Chaturvedi et al., 2012)." I thought that SSP5-8.5 is the newest experiment under CMIP6. How can be the ref so old? Maybe a typo?

**Response:** Thanks for the careful checking and for drawing our attention to this typing error. Chaturvedi et al. use the older, but similar, RCP-8.5 scenario from CMIP5. We corrected the error in the revised manuscript.

Ln 107-108: about the thermodynamics vs dynamics add as ref D'Agostino et al., 2019 and 2020.

**Response:** We added the proposed reference D'Agostino et al. 2019 referring to the Northern Hemisphere since we think the explanation of the dynamic and thermodynamic component of the moisture budget is a valuable addition in this context. The additional information (underlined) is included as followed:

"This trend is expected to be the consequence of the warming of the Indian Ocean enhancing atmospheric moisture content and thus moisture flux convergence arising from changes in moisture which generally follow the Clausius-Clapeyron relation (Cherchi et al.,2011; Seth et al., 2013; Mei et al., 2015; Sooraj et al., 2015; Agostino et al., 2019). This so called thermodynamic effect dominates over the dynamic effect which refers to weaker winds and a reduced monsoon circulation due to a weakened Walker circulation and an expected decrease of rainfall (Vecchi et al., 2006; Mei et al., 2015; Sooraj et al., 2015; Agostino et al.,2019)."

Ln 111: "The uncertain role of . . ." Missing something here.

**Response:** Removed.

Ln 126: "67.5°0'0"E - 98°0'0"E and latitude 6°0'0"N-36°0'0"N". I do not think you need coordinates in minutes and seconds here.

**Response:** We agree and thus, minutes and seconds have been removed in the revised manuscript.

Ln 250-253: refer to aforementioned studies about thermodynamics vs dynamics.

**Response:** As above, we added the references of D'Agostino et al., 2019 since we think the quantification of the dynamic and the thermodynamic component of moisture budget is a valuable addition here.

"Agostino et al. (2019) quantified the increase of the thermodynamic component of the moisture budget for the Indian monsoon with about 0.7mm/day and the decrease of the dynamic component with 0.4mm/day using nine CMIP-5 models in RCP-8.5 determining the positive sign of the change in monsoon rainfall ( Agostino et al. 2019, Sooraj et al. 2015)."

Ln 253: linear -> linearly

**Response:** Corrected.

Ln 213: Discussion. . . and Conclusions?

**Response:** For clarification, we separated the Conclusion paragraph.

Ln 283: "In this study, we used 32 CMIP-6 models to analyse the Indian summer monsoon's response to climate change." I would not repeat "in this study. . .".

**Response:** Removed.

---

## Author Response (AR2)

**Dear Dr Kirk-Davidoff,**

Thank you for handling the review process of our manuscript. Please find below our detailed response (In green) to the request (In black) of minor revisions of our manuscript.

Best wishes,
Anders Levermann

**Reviewer #1**
The manuscript has been significantly improved. However, there are still some parts not clear or partially incorrect. Some changes are still needed.

**We thank the reviewer for their detailed feedback regarding the connection of the modulation of the Hadley cell and changes in monsoon rainfall. We adapted the paragraph In accordance with the valuable propositions. Detailed comments can be found below in green, changes in the manuscript are underlined.**

Ln 59-62: " The resulting decrease in the land-sea thermal gradient over South Asia opposes the pressure gradient driving the Hadley circulation and consequently subdues the Hadley circulation. Since the Hadley system is responsible for transporting the rainfall to the subcontinent, this is accompanied by a reduction of the rainfall amount during the summer period as observed since the 1950s (Roxy et al., 2015)."

This is too general statement.

First, the Hadley Circulation (HC) that is "transporting" precipitation from Indian Ocean to Asian Continent in JJA, is the cross-equatorial Hadley Circulation, i.e. the strongest one in the Southern Hemisphere (SH). Second, the advection of moist static energy and moisture within the SH HC depends on atmospheric net energy input contrast (and not simply thermal contrasts: that is why the orbital forcing is much stronger driver of monsoons than GHG. See D'Agostino et al., 2019, 2020 where you have mid Holocene and rcp8.5 net energy input maps!).

The observed reduced Indian rainfall since 1950 as suggested by Roxy et al., might be due to:
decadal-to-multidecadal-centennial variability
SST/ surface fluxes over the Indian Ocean
compensating effect between GHG and Aerosols over land but more likely…
… (and this has been not explicitly mentioned) to a squeezing of the rain belt around the equator. The rain belt under GHG forcing and global warming is expected to be narrower and precipitation more intense (Byrne et al., 2015). So the reduced precipitation over the Indian continent might be simply due to a shift of the rainfall pattern from land to ocean (D'Agostino et al., 2019, 2020).

As also Roxy et al., 2015 said this somewhere "Furthermore, this enhanced upward motion over the ocean is compensated by subsidence of air over the subcontinent (10–20°N), inhibiting convection over the landmass and drying the region, through a modulation of the local Hadley cell (Fig. 5a). This suggests that though the warming ocean engenders enhanced local rainfall due to increased moisture availability, it weakens the monsoon Hadley circulation and reduces the rainfall over the land, ultimately building up a competition among the land and ocean rainfall in the South Asian monsoon domain."

Try to rephrase here please.

Include Byrne, M. P., & Schneider, T. (2016). Narrowing of the ITCZ in a warming climate: Physical mechanisms. Geophysical Research Letters, 43(21), 11-350.

Following the propositions of including more details, we decided to rephrase here and included the reference of Byrne and Schneider (2016): "The Indian ocean warming has been linked to anomalies in the lower and upper troposphere due to enhanced latent heat uplift resulting from convection over the ocean (Danielsen, 1993; Dai et al., 2013). The warming of the Indian ocean could intensify the convection over the ocean which is compensated by the subsidence of air masses over land. By preventing the convection over the subcontinent,

the Hadley cell is modulated in such a way that a drying trend over the region is introduced (Roxy et al.,2015). Another significant aspect contributing to the rainfall decrease is discussed to be the narrowing of the ITCZ and correspondingly, the decrease of the associated belt of intense rainfall (Byrne and Schneider, 2016)

Ln 85: "… reduced monsoon circulation due to a weakened Walker circulation and an expected decrease of rainfall …"

Incorrect. Replace "Walker circulation" with "tropical overturning".

Replaced.

**Reviewer #2:**

Publish as It Is.